# Modern Applications of 3D Printing: The Case of an Artificial Ear Splint Model

**DOI:** 10.3390/mps4030054

**Published:** 2021-08-06

**Authors:** Athanasios Argyropoulos, Pantelis N. Botsaris

**Affiliations:** Laboratory of Mechanical Design, Department of Production Engineering and Management, Democritus University of Thrace, Vasilissis Sofias 12, 67100 Xanthi, Greece; panmpots@pme.duth.gr

**Keywords:** stick-out protruding ears, ear aligners, PCL100, Z-Flex, 3D printing

## Abstract

Three-dimensional (3D) printing is a leading manufacturing technique in the medical field. The constantly improving quality of 3D printers has revolutionized the approach to new challenges in medicine for a wide range of applications including otoplasty, medical devices, and tissue engineering. The aim of this study is to provide a comprehensive overview of an artificial ear splint model applied to the human auricle for the treatment of stick-out protruding ears. The deformity of stick-out protruding ears remains a significant challenge, where the complex and distinctive shape preservation are key factors. To address this challenge, we have developed a protocol that involves photogrammetry techniques, reverse engineering technologies, a smart prototype design, and 3D printing processes. Specifically, we fabricated a 3D printed ear splint model via fused deposition modelling (FDM) technology by testing two materials, a thermoplastic polyester elastomer material (Z-Flex) and polycaprolactone (PCL 100). Our strategy affords a custom-made and patient-specific artificial ear aligner with mechanical properties that ensures sufficient preservation of the auricular shape by applying a force on the helix and antihelix and enables the ears to pin back to the head.

## 1. Introduction

3D printing in healthcare is a promising technique for future therapy of auricular cartilage congenital deformities. According to the classification of congenital auricular anomalies, stick-out protruding ears are regarded as a “deformational auricular anomaly” [1,2]. This anomaly is the most common ear deformity, with an incidence rate of 5% worldwide, and is a hereditary disease [2]. More accurately, it derives from a combination of imperfections of the antihelix side and the concha, which makes the ears stick out more than 2 cm and usually includes an underdeveloped antihelical fold and increased cephaloauricular distances [1,2]. According to the literature, stick-out ears is a deformity that affects both sexes equally, causing consequent psychological troubles throughout life, without affecting hearing, and leading to reduced confidence and poor social interaction. The impact of bullying and teasing are motivating factors for many children or adolescents to seek surgical correction [3]. The methods that exist for stick-out ear treatment includes both surgical and non-invasive techniques. The most commonly requested facial plastic surgery procedure worldwide is the otoplasty technique. Most otolaryngologists advise their patients to undergo ear surgery between the ages of 4 and 14 because the ear cartilage is not adequately developed; approximately 85% of auricular growth has been completed by three years of age. The main target of the surgical correction of stick-out ears is to reshape part of the cartilage, allowing ears to lie closer to the side of the head and reducing the cephaloauricular distances [2,3]. On the other hand, Earbuddies constitute a non-invasive method for stick-out ear correction in newborns. This technique involves a splint that is positioned in the helix side that is fastened with a tape, and the pressure of the splint restores the shape of auricular cartilage. Gradually, an accurate ear shape is created, resulting in a well-defined antihelical fold, as described in Figure 1 [4]. Nevertheless, there are still some problems with the existing surgical methods, with the most common one being the fact that there is no permanent solution. In fact, otoplasty techniques face serious problems, such as the high cost, approximately 5000 euro per case, while after completing the treatment, in some cases there are still aesthetic problems involving bleeding, hematoma, and pain [2,3].

With the respect to the latter, we have already published a non-invasive method for the correction of stick-out ears to the ICMMEN 2020 conference and building upon our prior work and motivated by the recent contribution of additive manufacturing processes to the medical field, we tried to improve our design parameters and test different materials [5]. Specifically, the ear model was taken by 3D scanning techniques, and reverse engineering processes were then used for the integration of complex surfaces from the human auricular. Through computer aided design (CAD), the patients can benefit from a cost-effective customizable artificial ear aligner.

## 2. Experimental Design

The fabrication of the artificial ear splint follows an innovative protocol, involving photogrammetric techniques, reverse engineering technologies, smart design, and 3D printing processes. Specifically, the photogrammetry is used as a 3D scanning technology for providing 3D surface data and accurate digital models, by scanning in our case a human ear, resulting in a digital ear model. There is always the risk of misalignment, but it can be prevented by taking multiple photos of adjacent areas and by using 3D imaging equipment and distinct software in order to obtain sufficient data and eliminate the noise from the model [5,6,7,8,9]. The digital ear model is composed of the data generated by using reverse engineering software. These data initially consisted of a set of points that were not connected and represented the ear surfaces. A reverse engineering program, like Geomagic, is a software that combine all these points together and provides digital models and assemblies from natural objects in a CAD form. The representation of the digital ear model may not end up with the exact dimensions because of the absence of scans and the complexity of the human ear [8]. However, this is an auricular shape comparison problem that can be faced with use of Meshlab, a program that provides various functions for editing and defining project points. Particularly, we overcome this problem by measuring the actual distances of three points between the helix and the head. Through a computer aided design (CAD) process, specific-patient and custom-made ear aligners can be generated. The new smart design involves the addition of material to the helix and antihelix sides in order to apply a more distributed force in comparison to our previous design, Figure 2. 3D printing of the artificial ear splint allows for the use of the fused deposition material technique and stereolithography to test various materials [5,6,7,8,9].

### 2.1. Materials

The two materials tested during this research were Z-Flex and Polycaprolactone (PCL100).
Z-Flex is an elastomeric and thermoplastic material that is manufactured by Zortrax in Poland. It was the first choice for our experimental plan because of its elasticity, mechanical properties, and the fact that it is a nontoxic filament [10]. After 6 h and 5 min, the ear splint made from Z-Flex was printed properly, weighting only 15 g, with dimensions according to the patient’s ear, 79.36 mm in length, 34.01 mm in width, and 44.05 mm in height.Polycaprolactone 100, known as PCL 100, is one of the most widely used materials in universities for medical research. Its flexibility and biocompatibility are the key factors for its selection as the 2nd material in this research study [11]. The printing time for the PCL 100 ear splint was 3 h and 39 min, and its weight was 14 g, while the dimensions were the same as those of the Z-Flex model.

It should be emphasized that the most important factors for a successful print are the nozzle and bed temperature and the print speed of the extruder, while the printing time for PCL was significantly less than that of Z-Flex, almost half. Print recommendations and the printing time according to the materials data sheets and our tests are described in Table 1 and Figure 3, respectively.

### 2.2. Equipment

The most well-known 3D printing technique is the fused deposition modeling (FDM) material extrusion technology, and our 3D-printer model belonged to this category. The 3d printer is developed by the Zortrax in Poland and it is known as the Zortrax M200. The M200 comes with a Hotend V2 that is adjusted to this model by the company, and it was changed to a Hotend V3 in order to execute our experiment, based on the company’s literature.

## 3. Procedure

In this study, the manufacturing process of the artificial ear splint was tested using two materials via fused deposition material technology, requiring a five stages procedure, which is described in Figure 4.

### 3.1. 3D-Scanning. 00:10 Hours

The whole scan procedure of the auricle can be done by using a photographic instrument, such as a smartphone. The scan should involve a 180-degree view of the human ear, start at the front side of ear and the tragus, then continue to the helix and antihelix sides and end at the back of the auricle beyond the connection to the head.Selecting of the appropriate place for scanning the patients’ ear.Exporting the digital ear model file.

### 3.2. Reverse Engineering. 01:30 Hours

The process of getting from a digital model to a CAD model requires two steps:Importing all these data from the scan photos to Geomagic and exporting a .obj file.Using tools in the Meshlab program to set up the scale of the digital ear model and measure the actual distances of three points between the ear and the head. Specifically, the distance between the top point of the helix and the head was 10 mm, the middle point was 16 mm, and the point from the lobule to the head was 19 mm [12].

### 3.3. Computer Aided Design. 00:30 Hours

The artificial ear splint CAD model was designed in the program Solidworks. Start designing the ear splint model by:Appling the delete face function on all the surfaces that are around the ear.Closing the holes by surface filling.Converting to mesh body.Increasing the mesh refinement to fine, so as to increase the corresponding triangles in the STL file.

The CAD file that should emerge from the process is the final design of the ear splint.

### 3.4. Conversion to STL & STL File Manipulation. 00:10 Hours

Regarding the STL manipulation, the whole process was executed in the Zortrax platform, the z-suite software. Start the procedure by:
1.Exporting the file from SLDPRT to STL.2.Using different custom profiles to set up the appropriate settings for both materials, Z-Flex and PCL 100.3.Z-Flex profile:
Nozzle diameter and layer thicknesses are 0.4 mm and 0.09 mm, respectively.Extrusion and platform temperatures are 230 °C and 60 °C, respectively.Extruder speed is 20 mm/s and the distance is 2.5 mm.Offset is 0.5.4.PCL 100 profile:
Nozzle diameter and layer thicknesses are 0.4 mm and 0.09 mm, respectively.Extrusion and platform temperatures are 130 °C and 30 °C, respectively.Extruder speed is 20 mm/s and the distance is 1.0 mm.Offset is 0.5.5.Placing the supports manually on both profiles.6.Exporting the gcode file and inserting it into the specific SD card.

### 3.5. 3D Printer Setup. 00:10 Hours

Once the STL file has been created, it should be inserted into a 3D Printer. Four tasks should be performed before pressing start printing:Heat the platform.Heat the extruder.Load material.Select the gcode file to print.Start the process.

### 3.6. Application. 01:30 Hours

Once the printing process has finished:Remove the printed model from the platform.Carefully remove all the supports using specific tools.

The surface of the printed model is 1 mm thick so it is easy to break. This helps facilitate the creation of a hole on the model, as described in Figure A1.

After completing each step of the procedure and applying both artificial splints on to the patient’s ear, it should look like Figure 5.

Figure 5 represents the artificial ear splint design being placed on a normal ear of a healthy individual as the overall goal of this project was to introduce the concept. It is true that in order to correct the shape of the ear, a second model of the artificial ear splint is required. This second design will essentially represent the ear in its corrected final position, and this will be used to correct the protruding ears. However, such analysis is ongoing, and data are not presented in this study.

## 4. Expected Results

We present a non-invasive treatment modality for the treatment of stick-out protruding ears by using a specific patient 3D printing artificial ear aligner. Its purpose is to bring the ears back to the head, according to the anatomical features of each person. By applying the artificial ear aligner on to the patient’s ear, our approach achieves substantial results. In particular, it is lightweight, only 15 g for the Z-Flex model and 14 g for the PCL 100, and because the fabricated ear splint corresponded to the digital model, both ear splints were fitted exactly to the patient’s ear. After testing, there were no complaints of discomfort, cartilage infection, or bleeding or scraping, which are usually encountered with other otoplasty techniques. It should be noted that it is possible for the patient to wear glasses over the aligner [6,9]. However, it is worth mentioning that the ear aligner from Z-Flex was tighter and rigid to the helix and antihelix sides compared to PCL 100. In addition, the Z-Flex was more fragile, and it failed more easily during the process of removing all the supports compared to the PCL 100.

## 5. Recommendations

Although our approach is a promising non-invasive technique for the correction of stick-out protruding ears, the artificial ear aligner was fabricated in the context of research, and it is still in the experimental stage. With respect to that, we propose future recommendations and next steps that should be taken for manufacturing an industrial product. First and foremost, a clinical trial should be executed with our prototype in patients of different ages, like babies and adults, so as to obtain sufficient data for the fruitfulness of our method. In addition, more 3D Printing techniques (like stereolithography) and materials (like flexible resin) should be tested to evaluate the characteristics of each artificial ear splint and a statistical analysis should be performed to evaluate their prospects.

## 6. Patents

This research was patented by the Hellenic Industrial Property Organization with patent number GR1009882B.

## Figures and Tables

**Figure 1 mps-04-00054-f001:**
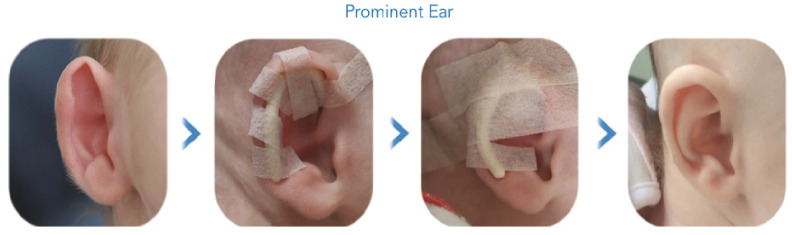
Non-invasive procedure of the Earbuddies method of treating stick-out ears [4]. Copyright 2021 EarBuddies^TM^.

**Figure 2 mps-04-00054-f002:**
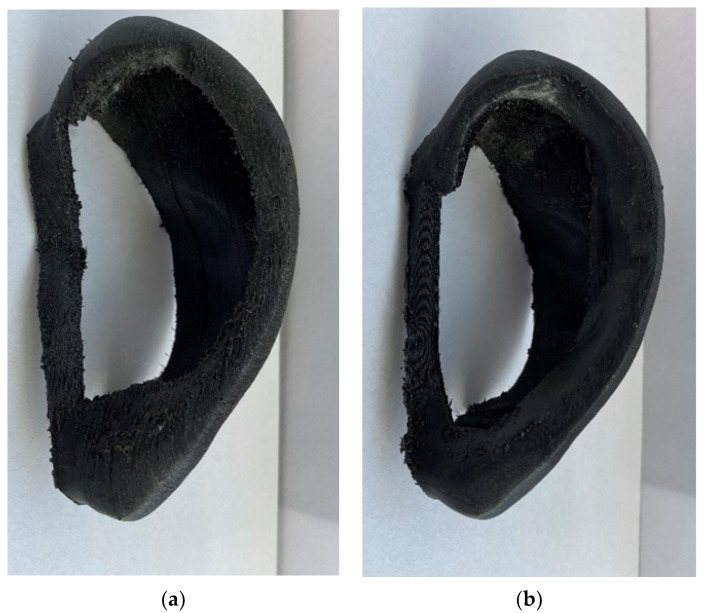
(**a**) Previous design of artificial ear splint. (**b**) Current design.

**Figure 3 mps-04-00054-f003:**
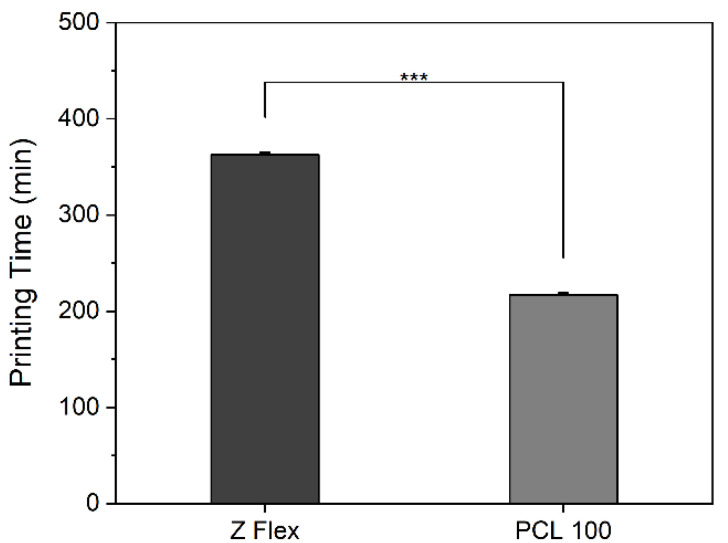
Printing time for the artificial ear splint from material Z Flex and PCL 100 (*** *p* ≤ 0.0001).

**Figure 4 mps-04-00054-f004:**
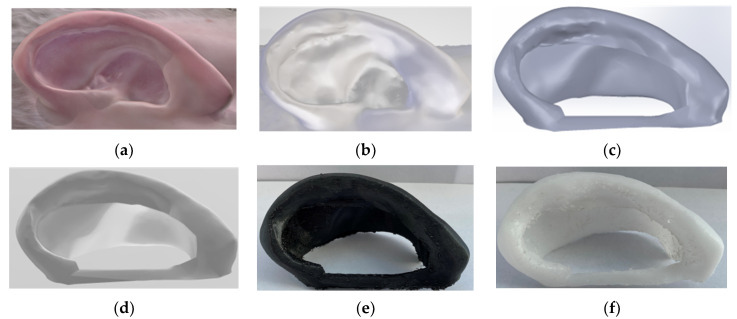
Demonstration of the manufacturing process of the artificial ear aligner and how it should look after each step (**a**) Ear model after scanning; (**b**) Digital ear model after applying reverse engineering technology; (**c**) Final design in Solidworks; (**d**) Final Design in STL file; (**e**) Artificial ear splint made by Z-Flex; (**f**) Artificial ear splint made by PCL 100.

**Figure 5 mps-04-00054-f005:**
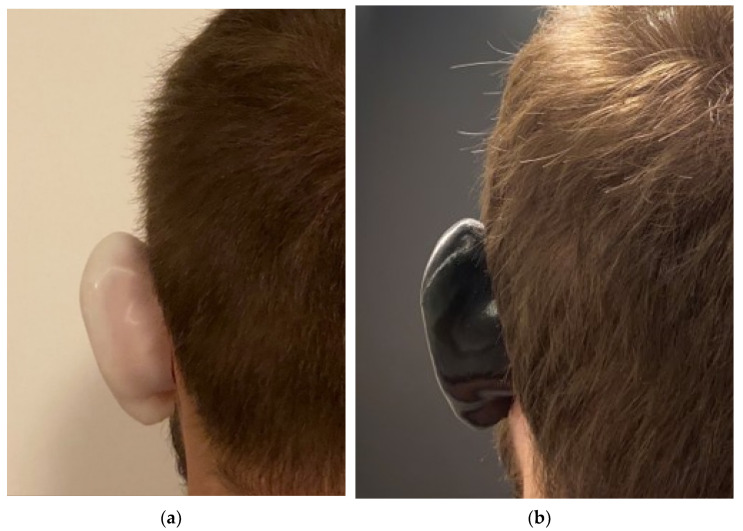
(**a**) Back View of artificial ear splint by PCL 100; (**b**) Back View of artificial ear splint by Z-Flex.

**Table 1 mps-04-00054-t001:** Print recommendations for Z-Flex and PCL100 [10,11].

Print Recommendations	Nozzle Temperature	Bed Temperature	Print Speed
Z-Flex	230 °C	60 °C	45 mm/s
Polycaprolactone 100	130–170 °C	30–45 °C	10–30 mm/s

## Data Availability

The data sets generated during and/or analyzed during the current study are available from the corresponding author on reasonable request.

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
