# Peer review of "Modern Applications of 3D Printing: The Case of an Artificial Ear Splint Model"

_mps, 2021, doi:10.3390/mps4030054_

Round 1
Reviewer 1 Report
Manuscript ID: mps-1269532
Title: Modern Applications of 3D Printing, The case 2 of an Artificial Ear Splint Model
Dear Editor and authors;
This study is research on development of a 3D printed ear splint for ‘stick out ears’. Two kinds of materials; polyester elastomer (z-flex) and polycaprolactone were used as print materials. Earbuddies has been widely used for correction of stick out ears. Therefore, if this new splint is going to replace the earbuddies, the printed splint should have more advantages over the earbuddies. I recommended the author to describe the advantages of this 3D printed splint; for example, cost aspect, compliance.
- [line 163 and 168] “ ~ layer thickness to 0.4 and 0.09” : Please indicate the unit.
- [line 188] As you know, the earbuddies are very soft to bend, easy to wear and affix to ears. However, the model has 1mm thickness to vulnerable to break. Is it safe when a baby does his/her daily activities? How to affix the splint to ears? With tape like earbuddies?
- Please describe mechanical properties of the scaffold. Tensile strength, flexibility, hardness, etc.
- [line 199] “ olny 15g for the Z-Flex ~” : on line 94, the Z flex weighs only 14 grams. Which one is the correct one?
- [line 203] “ is possible to wear myopic” : I think that “ ~ is possible to wear glasses” is more appropriate description.

Author Response
"Please see the attachment."

Reviewer 2 Report
introduction: stick out protruding ear is double. It is protruding ears or stick out ears.
Very interesting method to print a 3D splint for the treatment of protruding ears. I understand that it is just a presentation of the printing technique.
There are no results presented of the correction of protruding ears. So I do not get the conclusion that this is a good method to correct the deformity.
If you print the splint using photos of the prominent ear then the print has the shape of the prominent ear. It is not clear to me how this splint will correct the conchal bowl and create the antihelical fold. The shape of the ear will change after molding. Is the printed splint adjustable or do you have to print a new one?
The conclusion is not supported by the results. You can not conclude that this is a good method for molding protruding ear if you do not show the results of molding.
The technique is interesting, but might need some adjustments.
Author Response
"Please see the attachment."

Round 2
Reviewer 1 Report
Dear authors
The manuscript was well revised.
Thank you.
Reviewer 2 Report
Description is better, it can be interesting to see how it will develop further into a device that you can actually improve the auricular shape with.
Only comment:
Nevertheless, there are still some problems with the existing methods, with the most common one, the fact that there is no permanent solution.
This is incorrect. Molding methods, such as EarWell give a permanent solution of the problem.
